# A New Species of *Ampelovirus* Detected in *Persea lingue* (Ruiz & Pav.) Nees ex Kopp, a Common Tree of the Threatened Chilean Sclerophyll Forest

**Alan Zamorano** [1], **Camila Gamboa** [1], **Colombina Camilla** [1], **Francisca Beltrán** [1], **Carlos Magni** [2], **Suraj Vaswani** [2], **Eduardo Martínez-Herrera** [2] **and Nicola Fiore** [1,*]

[1] Facultad de Ciencias Agronómicas, Universidad de Chile, Santiago 8820808, Chile; agezac@uchile.cl (A.Z.); camila.gamboa@uchile.cl (C.G.); colocamilla.dlp@gmail.com (C.C.); fran.ibv24@gmail.com (F.B.)

[2] Facultad de Ciencias Forestales y de la Conservación de la Naturaleza, CESAF, Universidad de Chile, Santiago 8820808, Chile; crmagni@uchile.cl (C.M.); suraj@uchile.cl (S.V.); emartine@uchile.cl (E.M.-H.)

[*] Correspondence: nfiore@uchile.cl; Tel.: +56-2-2978-5960

**Abstract:** Biotic and abiotic stress seriously affects the development of plants, leading to the death of a significant number of plants in natural landscapes. Over the last twelve years, the central zone of Chile has been under an intensive drought, affecting the species inhabiting the Chilean Mediterranean forest, which is classified as a biological hotspot. In this context, our group started a small survey to monitor the presence of intracellular pathogens that may be infecting the plants, increasing the damage caused by the water deprivation. Several plants of one species, *Persea lingue*, showed symptoms of interveinal yellowing and leaf curling. RNA-seq analyses of one of these samples showed the presence of a long contig with high coverage only in symptomatic plants. Phylogenetic analysis and the molecular features indicated that it was a new species of the *Ampelovirus* genus. RT-PCR analyses determined the presence of the virus only in symptomatic plants of the same natural preserve.

**Keywords:** plant virus; forest; RNA-seq; *Persea lingue*; *Ampelovirus*

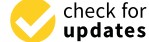



## 1. Introduction

*Ampelovirus* is a genus of plant virus belonging to the *Closteroviridae* family, members of which have long, helical, filamentous particles with positive-sense single-stranded RNA genomes [1]. The viral genomes of *Closteroviridae* contain two characteristic genomic blocks. The first is the replication-related gene block that includes the methyltransferase (Mtr) and helicase (Hel) domains encoded by open reading frame (ORF) 1a and the RNA-dependent RNA polymerase (RdRP) domain encoded by ORF1b. The second gene block encodes five proteins that function in virion assembly and transport, namely a small hydrophobic protein, a heat shock protein homolog (HSP70h), a ~60-kDa protein, the major capsid protein (CP) and a duplicated version of the latter (the minor capsid protein, CPm). Apart from the two conserved gene blocks, additional nonconserved genes encode proteins that vary in number, arrangement, function and origin within and between genera, with most having no detectable similarity with other viruses [1–3].

The Mediterranean-type climate forest of Chile is classified as a biological "hotspot" because of its high species diversity and high endemism [4]. The sclerophyllous forest (evergreen trees in valley) has been greatly impacted by anthropogenic activities, which has caused a considerable decrease in its natural vegetation cover, mainly due to its substitution by agricultural and forestry crops, urban expansion, overgrazing, unsustainable logging, and forest fires [5–8]. On the other hand, since 2010, central Chile (30–38° S) has been affected by an uninterrupted sequence of dry years provoking annual rainfall deficits (55%–75%) at regional level. In comparison with the instrumental historical period during

the last century, this hydric phenomenon has been recognized as a megadrought [9,10]. All these factors affect the recovery and regeneration capacity of trees [11]. Environmental factors that cause plant stress commonly predispose trees to forest pathogen attack. Plant viruses are rumored to lead to early senescence of trees, which is known to reduce the regeneration capacity of the host plants, and the juvenile metabolic vigor is lost prematurely. Thereby, virus-infected trees have a reduced potential for recovery from omnipresent abiotic stress conditions compared to non-infected trees [12]. Despite this, little attention is paid to plant viruses on forest trees. According to several studies performed in Europe, the lack of knowledge about the presence and frequency of occurrence of viral disease in forest trees leads to the impression that they are rare and therefore not important [13].

*Persea lingue* (Ruiz & Pav.) Ness (lingue) is an endemic tree growing in temperate forests of Chile and Argentina [14]. According to The IUCN Red List of Threatened Species in 2021, *P. lingue* is listed as Least Concern [15], but from parallel 35° to the north, it is considered vulnerable VU [16]. In Chile it has a wide geographical distribution, from Valparaiso (33°02′10″ S 71°37.8′ O) to Chiloé (42°28′20″ S 73°46′ O), from central valley to mountain with an altitudinal limit at 900 m above sea level [17]. In the central valley of Chile, *P. lingue* is found mixed with *Peumus boldus* Molina (boldo) and *Cryptocarya alba* (Molina) Looser (peumo), among other sclerophyllous species [18]. It has been highly exploited in the past for the excellent quality of its wood used in the manufacture of fine furniture, using the husk in leather tanning due to its high tannin content [19], and because its leaves have also been used in relation to potential antibacterial activity [20,21]. It also plays an ecosystemic role as a cavity tree, considered as a critical habitat for a range of species, from invertebrates through birds and mammals [22].

During the development of a Corporación Nacional Forestal project (CONAF FIBN-003-2020), we searched for different plants showing characteristic virus-like symptoms such as mosaics, chlorotic spots and mottling, in sclerophyll forest in the central zone of Chile. Among them, several individuals of *Persea lingue*, showed interveinal yellowing and chlorotic spots. In this work, we identified a novel species of the genus *Ampelovirus* associated with symptomatic plants using an RNA-seq approach to further associate the presence of the virus only with plants showing the described symptoms.

## 2. Materials and Methods

### 2.1. Plant Materials

All the samples were collected in Reserva Natural Privada Altos de Cantillana (33°58′ S–70°58′ W), located in the Metropolitan Region of Chile. For RNA-seq analysis, four branches with leaves were collected from a plant showing symptoms of leaf rolling, chlorotic mottling and/or interveinal yellowing (Figure 1). A second sampling was taken to obtain plant material for RT-PCR analyses. We sampled plants showing the symptoms described above and plants without symptoms, if available, as a control. Plants collected were mostly adults, presenting one main stem, but other plants of different ages presented bushy growth. In both situations, the sample collection was designed to obtain tissues from the entire canopy. The collected plants were distributed in different areas of the preserve, mostly located near water ravines.

### 2.2. RNA Extraction and Genome Sequencing Analyses

As the symptoms were similar among the plants, a symptomatic plant showing severe interveinal chlorosis and leaf rolling (Figure 1a) was subjected to RNA-seq. Total RNA was extracted using a Spectrum Total plant RNA extraction kit (Sigma Aldrich, Darmstadt, Germany) according to manufacturer's instruction. Library construction was performed using TruSeq Stranded Total RNA with a Ribo-Zero Plant kit. Paired-ends sequencing of 150 bp reads was performed using the NovaSeq6000 platform (Psomagen Inc., Rockville, MD, USA). Raw reads were trimmed and assembled using a CLC Genomics Workbench, v24.0.1. Viral contigs were identified using the tBLASTx tool in the CLC Genomics Workbench, creating a local database containing all the contigs obtained from *de novo* assembly

and using as a query a customized sequence list including the full genome of at least one species from each viral genus. In addition, the contig was subjected to the regular Blastx analysis in the National Center for Biotechnology Information (NCBI) database. Further analysis through reverse transcription-PCR (RT-PCR) was applied to the viral RNA considering four separated genes: RdRp, HSP70-like, CP and minor CP. Four sets of primer pairs, one for each gene, were designed based on the contig of the virus, amplifying fragments of ~400 bp in size (Table S1). Viral sequence ends were completed using a 5′/3′ RACE kit (Sigma Aldrich, Darmstadt, Germany) according to manufacturer's instruction. Partial genomic regions known to be conserved among *Ampelovirus* species were verified by the amplification using primers designed on the basis of the NGS-obtained sequence. Each fragment obtained was cloned using pGEM-T easy vector (Promega). Amino acid sequence alignments were performed using BioEdit [23] and MEGA v7.0 [24]. Phylogenetic analyses were performed using the maximum likelihood method by MEGA v7.0.

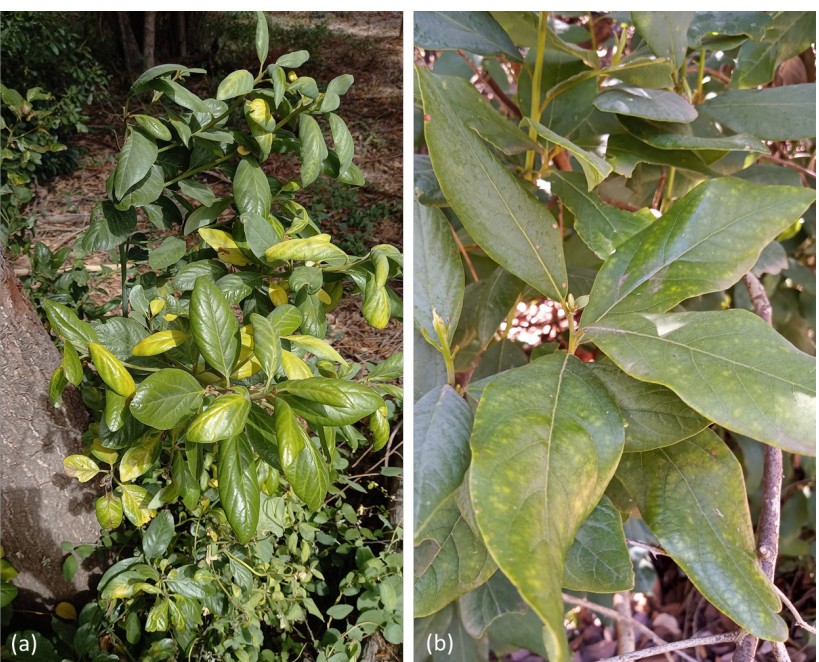

**Figure 1.** Symptoms observed in infected plants. (**a**) Interveinal chlorosis and leaf deformation; (**b**) chlorotic mottling of leaves.

### 2.3. RT-PCR Detection

RT-PCR was used to detect the virus in 20 lingue plants collected during a second sampling step. Two primers pairs were designed based on the sequences obtained by NGS and used for detection (LingAmpF1 5′-GCCAATTTAGCGGATGTTAATG-3′/LingAmpR1 5′-CAGCAGTGAAGTTGTCCATAG 3′) that amplified 414 bp from the CP gene sequence and (LingAmp-F3 5′-TTAAGACCAGGCTCGAGAATAA/LingAmp-R3 5′-GCTGTAGTGAGA-GGTGCTATAA) designed for the amplification of 434 bp of the minor CP ORF. PCR amplification conditions were initial denaturation of 1 min × 94 °C followed by 35 cycles of (94 °C × 20 s; 59 °C × 30 s; 72 °C × 30 s) and a final elongation of 72 °C × 5 min. Positive samples were cloned and sequenced to verify the correct amplification of the new *Ampelovirus*. The amplification product of LingAmp-F3/R3 primers from all the plants were sequenced, in silico translated, and aligned against the amino acid sequence of the partial CP gene of other *Closteroviridae* species, including the LinAV-1 sequence obtained by NGS.

## 3. Results

### 3.1. Identification and Molecular Characterization of the New Virus in Lingue

After next-generation sequencing, a total of 58,021.802 paired reads were assembled *de novo* into 61.765 contigs. A unique long contig of 13,989 bp and a total coverage of 132×, matched with members of *Closteroviridae* family, showing the highest matches with members of the *Ampelovirus* genus. The four viral genes (RdRp, HSP70-like, CP and minor CP) were amplified, sequenced and aligned against the complete sequence, matching all the fragments with 100% of nucleotide identity with the genome of the virus reaching a total of 14,028 nucleotides (Genbank accession number OQ805851). The complete genome of the identified virus is 14,028 nt in length, with nine predicted ORFs (Figure 2). The genome organization showed the typical structure of other members of the genus *Ampelovirus* [1,25]. The replication genes block, ORF1 and ORF2, presents a separated expression with a first protein of 1378 Aa-long with a viral methyltransferase conserved domain (pfam01660) located in positions 69 to 378 and a helicase domain (Pfam 01443) located in the C-terminal region of the encoded protein (positions 1084 to 1334). The HEL domain shows the six conserved amino acid motifs present in other *Ampelovirus* species (Figure S1a). ORF2 encodes a putative 538-amino acid peptide identified as the RNA dependent RNA polymerase (RdRp; pfam 00978). The predicted amino acid sequence contains the eight conserved motifs present in all RdRp of positive-strand RNA viruses (Figure S1b). The conserved ampelovirus p6a protein is coded in ORF3 showing a predicted molecular mass of 5.8 kDa (50 aa) and showing a putative transmembrane helix, by in silico analysis. The predicted protein encoded by ORF4 contains a molecular chaperone conserved domain, resembling heat shock protein 70 (pfam 00012), with 540 aa. ORF5 encodes a putative protein of 481 amino acids (55 kDa) that matches a previously identified heat shock protein 90 from Citrus ampelovirus 2 [3], but no conserved domains were identified using different online tools. Capsid protein (CP) and minor CP are encoded in ORF6 and ORF7 (312 and 467 amino acids, respectively), both showing the canonical *Closteroviridae* family capsid protein domain (pfam01785). ORF6 shows a higher score and showed the conserved motifs of the major CP identified in other members of the *Closteroviridae* family (Figure S1c), and ORF7 codes for a protein known as minor CP or CP duplicate. These two CP proteins, together with p6a, HSP70 and HSP90, comprise the quintuple gene block (QGB), previously reported as conserved in other viruses from the *Closteroviridae* family [1,25]. ORF8 is located downstream of the QGB, encoding a putative protein of 19.3 kDa. These proteins have no significant similarities with any proteins in the available databases that agree with the demarcation criteria of the *Closteroviridae* family, differing by more than 25% [1,25]. However, the usual functions of the proteins encoded by ORF8 and 9 of other members of the family *Closteroviridae* have been associated with silencing suppression and viral movement [26].

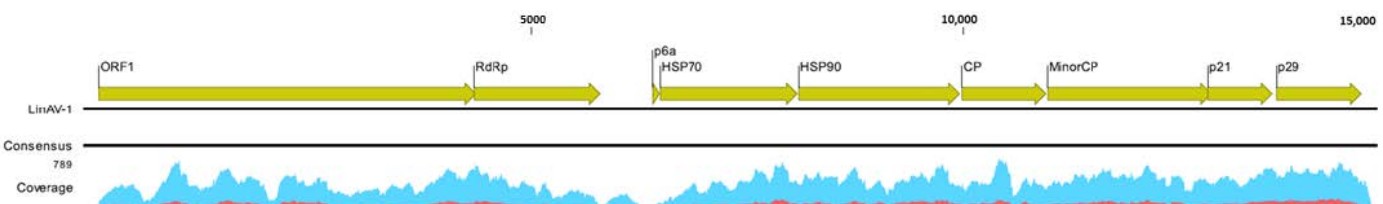

**Figure 2.** Genomic organization of the virus found in lingue.

### 3.2. Phylogenetic Relationships of the New Virus in the Closteroviridae Family

We performed comparisons of four taxonomically significant gene products commonly used for *Closteroviridae* members: ORF1, RdRp, HSP70 and CP proteins. Table 1 shows the amino acid sequence identities observed when the sequence of these genes was compared against other members of *Ampelovirus*, showing differences higher than 25% in all cases. Phylogenetic analyses of the amino acid sequences of the four genes mentioned

above showed that the new virus detected in lingue clustered with other ampeloviruses, confirming its classification as a member of this genus (Figure 3). It was noticed that the lingue virus is closely associated with *Citrus ampelovirus 2,* but with significant sequence differences that allow us to confirm its assignment as a distinct viral species belonging to the genus *Ampelovirus*, tentatively named *Lingue ampelovirus 1* (LinAV-1).

**Table 1.** Amino acid identity of the four taxonomically significant proteins of the new virus in lingue against members of *Ampelovirus* genus.

| Virus | ORF1 | RdRp | HSP70 | CP |
| --- | --- | --- | --- | --- |
| CaAV-1 | 42.30% | 60.80% | 39.50% | 24.10% |
| CaAV-2 | 39.00% | 59.30% | 49.90% | 47.70% |
| GLRaV-1 | 14.90% | 39.80% | 43.30% | 26.40% |
| GLRaV-13 | 14.90% | 36.10% | 40.50% | 24.40% |
| GLRaV-3 | 15.30% | 37.00% | 42.70% | 39.50% |
| LChV-2 | 19.00% | 42.90% | 30.40% | 10.60% |
| PAV-1 | 45.10% | 59.60% | 38.00% | 22.50% |
| PMWaV-2 | 16.10% | 33.40% | 39.50% | 24.90% |
| YaV1 | 25.40% | 44.20% | 30.90% | 14.60% |

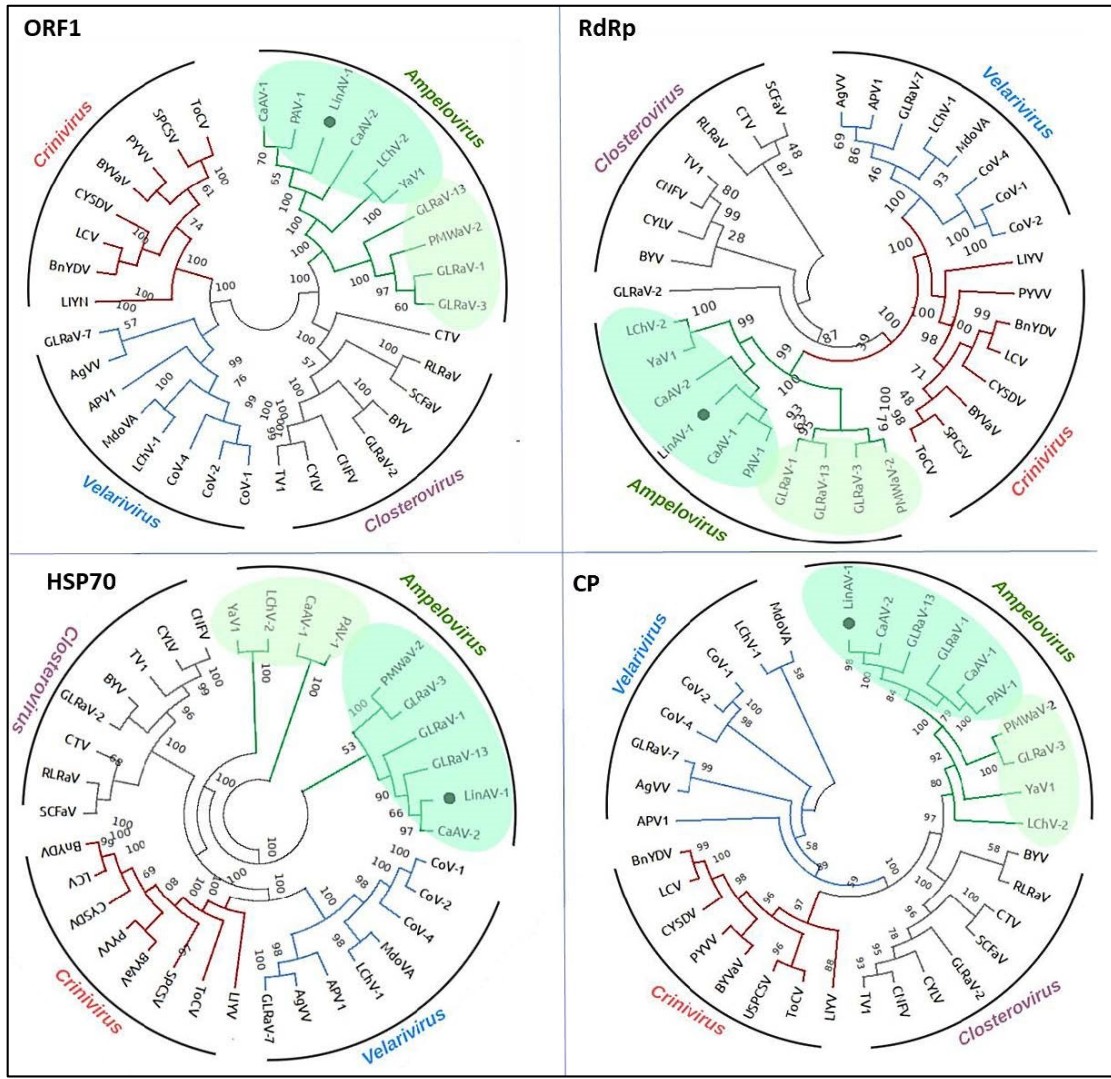

**Figure 3.** Phylogenetic relationships of the new virus in lingue among members of the *Closteroviridae* family, calculated using the four taxonomically significant genes.

*3.3. Survey of LinAV*

We monitored the plants located near the authorized trails of the preserve, covering an approximate area of 15 hectares to collect leaf samples from lingue specimens showing symptoms putatively associated to the presence of the virus. Among them, we observed samples with severe leaf rolling, in addition to the yellow spots described above, that were later tested positive for the presence of the virus. A total of 13 out of the 20 plants collected, tested positive for LinAV-1, showing that the virus is widespread in the lingue plants of the preserve. The amplification product of LingAmp-F3/R3 primers showed very few variations among lingue samples (Figure S2a), which was confirmed by a phylogenetic tree constructed using a maximum parsimony algorithm (Figure S2b).

## 4. Discussion

Owing to the intense megadrought affecting the central region of Chile, the sclerophyll forest is under severe threat [27], which is one of the most relevant drivers of this research, and there has been a concentrated effort towards restoration of the ecosystem. To date, this restoration has not been as effective as expected, possibly due to natural factors like seed maturation or anthropogenic factors like intense forest fires [28] or fragmentation of the forest, which may lead to a constantly increasing endogamy, as observed in other ecosystems of the country [29,30]. Now, the identification of viral pathogens in sclerophyll forest species appears as a potential new agent that may impair the efficiency of restoration. Even when there is no evidence of seed transmission in species belonging to the *Ampelovirus* genus, the presence of these viruses has previously been associated with alterations in berry maturation in grapevines infected with *Grapevine leafroll-associated virus 3* [31] or sweet cherry development in plants infected by *Little cherry virus 2* [32]. The impact of the presence of LinAV-1 in lingue fruit and seed quality has not been evaluated yet, but the alterations observed in the leaves are usually associated with alterations in photosynthetic performance, leading to physiological detriment of the plant, as has previously been observed in grapevine [33].

In addition to the survey of the virus in Alto de Cantillana preserve, we are focusing on the identification of a potential insect vector that may be associated with the spread of the pathogen. Previous studies indicate that *Ampelovirus* can be transmitted by several species of the Coccidae and Diaspididae families [25,34]. In our survey, we observed that most of the specimens testing positive for LinAV-1 were infested with a native species of armored scale *Abgrallapsis latastei* (Cokerell); therefore, the association of this scale with the virus spread is under analysis.

With the identification of the virus and the development of a diagnostic tool, we plan to extend the survey to other locations where we find sclerophyll forest in the central region of the country, in order to select healthy material that can present better rates of seed germination, increasing the success of restoration projects. This takes on greater relevance considering that, within these naturally discontinuous forests, *Persea lingue* is classified as a vulnerable species.

To our knowledge, this is the first report of a virus infecting plant species of Chilean sclerophyll forest and, moreover, the first report of a virus infecting sclerophyll species worldwide. Under the ongoing project, we plan to analyze more species of the Chilean native forests with the main purpose of improving the efficiency of restoration projects, and therefore, slowing down the intense degradation of this unique ecosystem.

## 5. Conclusions

The megadrought that has been affecting the country for more than a decade has raised the alarm about the rapid decline of Chilean sclerophyllous forests. It is therefore necessary to combine efforts to slow down this decline and optimize restoration processes. For this reason, the identification and subsequent control of new pathogens that could be involved in a lower restoration success rate is fundamental. This work represents the first relevant result of the ongoing project, which encourages us to continue working, in order

to have more clarity on the phytosanitary status of the forest and to contribute, from our point of view, to the process of rescuing the species of the Chilean sclerophyllous forest.

**Supplementary Materials:** The following supporting information can be downloaded at: https://www.mdpi.com/article/10.3390/f14061257/s1. Figure S1: *Ampelovirus* conserved motifs found in ORF1 (a), RdRp (b) and CP (c)-deduced amino acid sequences from LinAV-1; Figure S2: Validation of the detection of LinAV-1 in plants collected along Cantillana preserve by the Alignment (a) and the phylogenetic tree constructed by maximum parsimony (b) of the deduced amino acid sequences of a partial region of CP gene. Table S1: Primers designed for the validation of in silico obtained sequence of LinAV-1.

**Author Contributions:** Conceptualization, A.Z., C.M., S.V., E.M.-H. and N.F.; methodology, A.Z., C.G., S.V. and N.F.; validation, A.Z., C.G., C.C. and N.F.; formal analysis, C.G. and C.C.; data curation, A.Z., F.B. and N.F.; writing, review and editing, A.Z., F.B., C.M., E.M.-H. and N.F. All authors have read and agreed to the published version of the manuscript.

**Funding:** This study was funded by Corporación Nacional Forestal (CONAF), Fondo de Investigación del Bosque Nativo (FIBN), MINAGRI, Chile, Project No 003/2020.

**Data Availability Statement:** Full viral genome sequence is available under Genbank accession number OQ805851.

**Acknowledgments:** The authors thank Reserva Natural Privada Altos de Cantillana (RNAC), especially Fernanda Romero, coordinator researcher of RNAC, for the continued support in carrying out this study and to Danilo Cepeda, who helped in the identification of putative insect vectors.

**Conflicts of Interest:** The authors declare no conflict of interest.

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
