# Peer review of "A New Species of Ampelovirus Detected in Persea lingue (Ruiz & Pav.) Nees ex Kopp, a Common Tree of the Threatened Chilean Sclerophyll Forest"

_forests, doi:10.3390/f14061257_

Round 1

Reviewer 1 Report

1.       Line 2: It is better to be called ‘a new species of Ampelovirus’ instead of ‘a new Ampelovirus’.

2.       Line 13: Better to remove ‘interaction between’.

3.       Line 19-20: It is inconsistent with the description in the text. RNA-seq was performed just on one sample, and the verification was carried out with the CP gene sequence.

4.       Line 82: Please describe the sampling in detail.

5.       Line 177: [25] should be removed.

6.       Line 203: “of” should be removed.

7.       English language needs some editing.

8.       The format of Reference should be check carefully, there are quite a few format differences among references.

9.       Did any other viral sequence be detected in the RNA-seq due to the sample was exposed outside.

The authors identified a new virus which infects persea lingue and leads to obvious symptoms, and developed a diagnostic tool for the virus detection. It is helpful for the protection of Chilean sclerophyll forest.

moderate editing of english language required.

Author Response

Please find attached a word file with our replies to your review.

Best regards

Reviewer 2 Report

The current study describes the discovery and genetic characterization of a novel Ampelovirus in the sclerophyll species lingue (Persea lingue). This is a novel finding of epidemiological importance for the Chilean sclerophyll forest, already suffering from plenty of biotic and abiotic stress factors. The characterisation of the novel virus is appropriately accomplished and issues aspects of the pathogen epidemiology like pathways of  spread. The manuscript surely merits publication to achieve further dissemination of these results.

The following comments may support the improvement of the manuscript.

l. 52: despite of this

M&M

l. 86: Do you mean that one sample from one plant was used for RNA-Seq? How many leaves were used for each sample? Please be more explanatory.

You will read later that some exchanges are needed between the M&M and Results sections.

Results

l. 123-139: In the section 3.1. there is redundant information from the M&M section. These are for example the sentences:

“Next-generation sequencing was performed on a plant that showed severe interveinal 123 chlorosis and leafrolling (Figure 1a). After trimming by quality …”

“The contig was subjected to the regular Blastx analysis in the National Center for 129 Biotechnology Information (NCBI) database, …”

“To verify the sequence of 131 the in silico identified viral RNA, reverse transcription-PCR (RT-PCR) was performed on 132 total RNA extracted from the infected lingue leaves, considering four separated genes: 133 RdRp, HSP70-like, CP and minor CP. Four sets of primer pairs were designed based on 134 the contig of the virus, amplifying fragments of ~400 bp in size. “

“RACE protocol was 137 followed to complete 5’ and 3’ sequences, …”

Please move this information to the M& M.

After removing this info, I think it makes sense to merge section 3.1 and 3.2 to one section about “identification and molecular characterization of a new virus”.

Section 3.2, l. 167/168: Please mention the species demarcation criteria in the genus Ampelovirus.

Section 3.4

l. 189-192: This info belongs to the M&M, please move it there.

Discussion

l. 203: “the sclerophyll 202 forest of ??? (is the a word missing?) is under a severe threat “

Author Response

(The authors gave the same response as above.)
